# Distribution Alignment for One-Shot Federated Learning via Optimal Transport

**Daniele Berardini** [1]  **Vito Paolo Pastore** [1 2]  **Vittorio Murino** [1 3]

## Abstract

One-Shot Federated Learning (OSFL) addresses extreme communication regimes in which clients interact with the server only once, amplifying the impact of heterogeneous client data distributions. In particular, the interaction of domain shift and label shift across clients induces misaligned feature representations that cannot be corrected through iterative optimization. Existing OSFL methods rely on distillation, server-side generation or ensemble-based aggregation, but assume aligned representations or address domain and label shift separately. We introduce SLOT-ALIGN (Single-round, Learning-free Optimal Transport Alignment), a geometry-aware feature harmonization framework for OSFL. SLOT-Align uses a shared frozen encoder to extract compact feature statistics, constructs a global reference via Bures–Wasserstein barycenters, and aligns local representations using closed-form geodesic optimal transport maps. The method is computationally efficient and can be combined with existing OSFL pipelines relying on frozen encoders without modifying their training procedures. Extensive experiments across multiple benchmarks, pretrained backbones, and OSFL methods show that SLOT-Align consistently improves accuracy and robustness under joint domain and label shift.

## 1. Introduction

Federated Learning (FL) (McMahan et al., 2017; Kairouz et al., 2021) enables collaborative model training across multiple clients while keeping data decentralized, motivated by privacy preservation and limited communication budgets. In this paradigm, each client $k$ holds data drawn from a local joint distribution $\mathbb{P}_k(x, y)$, and the goal is to learn, via a central server, a model that performs well across the population without directly sharing raw data. A fundamental challenge in FL arises from *statistical heterogeneity*, since client data are typically not independent and identically distributed (Li et al., 2020; Zhu et al., 2021).

In realistic deployments, heterogeneity manifests along multiple axes (Chen et al., 2025a). First, clients may observe different input distributions $\mathbb{P}_k(x)$ due to variations in acquisition conditions, environments, or sensing pipelines, a phenomenon commonly referred to as *domain shift*. Second, clients often exhibit different label marginals $\mathbb{P}_k(y)$, a form of heterogeneity commonly known as *label shift*, which arises naturally in decentralized settings with varying class proportions across clients. Together, domain shift and label shift induce client-specific posterior distributions $\mathbb{P}_k(y \mid x)$, which determine the class-posterior decision functions implicitly learned by local models during training. As a result, clients learn feature representations and class-posterior decision functions that are adapted to their local data distributions rather than to a shared global objective. When client contributions are aggregated at the server without explicit mechanisms to mitigate distributional discrepancies, these mismatches can propagate into the global model and degrade performance (Huang et al., 2024).

In conventional multi-round federated learning, such inconsistencies are mitigated through iterative client–server interaction. Repeated rounds of local optimization and aggregation allow the global model to progressively adapt to heterogeneous client distributions. This adaptation is achieved either through server-side optimization strategies that adjust the aggregation process (Karimireddy et al., 2020; Li et al., 2023a; Chen et al., 2023a; Li et al., 2023b), or through client-side regularization mechanisms that guide local updates using global model feedback and loss-level adjustments (Kim et al., 2022; Qu et al., 2022; Zhang et al., 2022b). These mechanisms enable partial correction of mismatched objectives and representations over time.

However, an increasing number of applications impose strict communication budgets or completely prohibit iterative interaction. This has motivated the study of *One-Shot Federated Learning* (OSFL), where each client is allowed to communicate with the server only once. Under this con-

---

[1]AI for Good (AIGO), Italian Institute of Technology, Genoa, Italy [2]MaLGa-DIBRIS, University of Genoa, Genoa, Italy [3]Department of Computer Science, University of Verona, Verona, Italy. Correspondence to: Daniele Berardini <daniele.berardini@iit.it>.

*Proceedings of the 43rd International Conference on Machine Learning*, Seoul, South Korea. PMLR 306, 2026. Copyright 2026 by the author(s).

straint, the server must construct a global model from a single exchange of client information, without the possibility of iterative learning or refinement, which amplifies the impact of distributional heterogeneity. Existing OSFL approaches, which typically attempt to address either label imbalance or domain shift, differ primarily in the type of information transmitted by clients and in the mechanism used by the server to construct a global model (Liu et al., 2025). Broadly, these methods rely on single-round model aggregation or ensembling (Talpini et al., 2025), on distillation-based transfer of model outputs or logits to the server (Li et al., 2021a; Gong et al., 2022; Song et al., 2023), or on generation-based strategies that synthesize proxy data for server-side training (Heinbaugh et al., 2023; Yang et al., 2024b; Zhang et al., 2024; Yang et al., 2024a; Chen et al., 2025b; Yang et al., 2025). More recent work has increasingly focused on statistics-based and representation-based OSFL strategies that leverage frozen, pretrained encoders and transmit lightweight feature statistics or parametric summaries (Guan et al., 2025; Beitollahi et al., 2025). Although effective in reducing communication and computation, these approaches implicitly assume that client feature representations are already well aligned. As a result, discrepancies induced by domain shift and amplified by label imbalance remain largely uncorrected in the one-shot setting.

In this work, we therefore focus on OSFL settings characterized by domain shift, with additional label shift across clients, and introduce SLOT-ALIGN[1] (Single-round Learning-free Optimal Transport Alignment), a geometry-aware feature alignment framework designed specifically for this regime. SLOT-Align, operating as a preprocessing step, computes compact first- and second-order statistics of local feature distributions induced by a shared frozen encoder; aggregates these statistics into a global reference distribution using Bures–Wasserstein barycenters; and constructs closed-form geodesic optimal transport maps that align local feature representations toward this reference. The entire procedure is *computationally efficient*, *training-free*, and can be *seamlessly combined with arbitrary OSFL algorithms* that rely on pretrained encoders. Our contributions are threefold:

1. We formalize OSFL as a distribution alignment problem under heterogeneous client distributions $\mathbb{P}_k(x, y)$, highlighting how the interaction of domain shift and label shift can limit the efficacy of one-shot approaches.

2. We propose SLOT-Align, a learning-free, optimal-transport-based alignment framework that harmonizes client feature distributions through a single exchange of compact statistics, without modifying downstream OSFL optimization procedures.

---

[1]Project page: https://github.com/daniebera/SLOT-Align

3. We conduct an extensive empirical evaluation across multiple datasets, pretrained backbones, and state-of-the-art OSFL methods, showing that SLOT-Align consistently improves performance and robustness under domain shift compounded by label shift.

## 2. Related Work

### 2.1. One-Shot Federated Learning

OSFL considers an extreme communication regime in which clients are allowed to interact with the server only once, precluding any iterative client–server optimization. This constraint is motivated by communication and privacy considerations and fundamentally differentiates OSFL from multi-round federated optimization. Early work, including one-shot extensions of FedAvg (McMahan et al., 2017; Guha et al., 2019), framed OSFL as a single-round model integration problem, enabling global inference or training without explicitly addressing client heterogeneity. The OSFL literature is commonly organized around the dominant server-side mechanism used to integrate client information, even though many methods combine multiple techniques, such as distillation, data synthesis, and ensembling.

Several OSFL methods adopt knowledge distillation as a central mechanism, where information learned by local models is transferred to a global model on the server. Model-level distillation approaches treat client models as teachers and train a global student using transferred predictions or logits, typically evaluated on auxiliary or public data (Lin et al., 2020; Li et al., 2021a; Gong et al., 2022). FedKT (Li et al., 2021a) aggregates multiple client teachers via server-side distillation, while FedKD (Gong et al., 2022) distills an ensemble of local models into a single global student. Other approaches extend this paradigm to data distillation, compressing client knowledge into synthetic samples or distilled representations for server-side training, as in DOSFL (Zhou et al., 2020) and FedD3 (Song et al., 2023). Although these approaches support model heterogeneity and can alleviate certain forms of label skew, they typically rely on auxiliary data or server-side optimization and address distributional heterogeneity only implicitly through the distillation process.

Complementary to distillation-centric approaches, several OSFL methods place data synthesis at the core of the server-side design, using learned generative models to approximate client data distributions and synthesize training samples on the server. FedDEO (Yang et al., 2024b), FedDISC (Yang et al., 2024a), and FedBiP (Chen et al., 2025b) leverage pretrained diffusion models to generate client-aligned samples, while FEDCVAE (Heinbaugh et al., 2023) and FedSD2C (Zhang et al., 2024) rely on locally trained variational autoencoders to reconstruct synthetic datasets on the server.

While effective under severe heterogeneity, these methods require additional server-side optimization and incur substantial computational and memory overhead due to generative model training or fine-tuning. Moreover, synthetic data generation may raise additional privacy considerations. In contrast, SLOT-Align operates directly on compact feature statistics and introduces no learning or data synthesis stages.

In parallel, ensemble-based OSFL methods construct a global predictor by combining local models or sub-models through static or adaptive weighting strategies. DENSE (Zhang et al., 2022a) couples ensembling with data-free distillation, while Co-Boosting (Dai et al., 2024) alternates between data generation and ensemble refinement. Other works, such as FENS (Allouah et al., 2024) and FuseFL (Tang et al., 2024), relax the strict one-shot constraint by replacing a single exchange with lightweight iterative aggregation schemes. Notably, these works argue that communication efficiency is more important than adherence to a strict single-exchange protocol, but they primarily focus on model-centric aggregation and refinement rather than explicit distribution alignment across heterogeneous client data.

Orthogonal to model-centric integration strategies, a growing line of work studies OSFL through statistics-based and representation-based aggregation. Rather than transmitting full models, clients communicate compact summaries such as feature statistics, class prototypes, or parametric approximations of local representations. This paradigm is particularly appealing in vision tasks, where strong pretrained encoders can be shared across clients. Recent work includes FedCGS (Guan et al., 2025), which aggregates global feature statistics extracted from pretrained backbones, and FedPFT (Beitollahi et al., 2025), which models client feature distributions parametrically and trains server-side classifiers using synthetic features. While these methods are lightweight and avoid heavy learning-based synthesis or local model refinement, they typically assume that client feature summaries are directly comparable and do not explicitly correct for distributional misalignment induced by domain shift compounded by label shift.

### 2.2. Optimal Transport for Distribution Alignment

Distribution alignment has been widely studied as a means to reconcile heterogeneous data sources in machine learning. Optimal transport (OT) provides a general framework for aligning probability distributions by accounting for the geometry of the underlying space and has been successfully applied in domain adaptation (Courty et al., 2016; 2017; Xu et al., 2020), multi-source learning (Montesuma & Mboula, 2021), and representation alignment (Singh & Jaggi, 2020). In these contexts, OT-based methods enable distribution-level alignment that explicitly accounts for discrepancies in

the underlying data geometry.

Despite these successes, the adoption of OT in federated and distributed learning remains limited. Existing approaches are inherently multi-round, relying on iterative optimization across clients and the server. FedOT (Farnia et al., 2022) addresses personalized federated learning by iteratively learning optimal transport maps and prediction models across clients under a multi-marginal optimal transport formulation. FedDaDiL (Castellon et al., 2024) studies decentralized multi-source domain adaptation through iterative computation of Wasserstein barycenters and repeated updates of client models. As a result, the applicability of OT-based ideas to OSFL remains largely unexplored in non-iterative settings.

Motivated by the lack of systematic integration of OT principles in OSFL, and drawing on established OT theory, SLOT-Align introduces a geometry-aware feature alignment mechanism designed as a lightweight, modular preprocessing step. The method operates on a single exchange of compact feature statistics and aligns client feature distributions prior to downstream OSFL methods, without requiring auxiliary data, trainable generators, or iterative optimization. Rather than replacing existing OSFL pipelines, SLOT-Align explicitly corrects mismatches in first- and second-order feature structure induced by heterogeneous $\mathbb{P}_k(x, y)$, which are typically left unaddressed by aggregation- or distillation-based integration strategies. This design enables SLOT-Align to be seamlessly combined with the growing body of downstream OSFL methods that rely on frozen encoders, without modifying their communication patterns or optimization procedures. As a result, SLOT-Align improves robustness in settings where domain shift is compounded by label shift, which systematically distorts client feature statistics and remains insufficiently addressed by existing OSFL techniques.

## 3. The Method: SLOT-ALIGN

In this section, we formalize the statistical setting of OSFL and then present SLOT-Align, a geometry-aware alignment framework for OSFL methods. A schematic overview of the proposed approach is provided in Fig. 1.

### 3.1. Statistical Setting and Notation

We consider a federated environment with $K$ clients. Each client $k$ holds a local dataset drawn independently from a client-specific joint distribution $P_k(x, y)$ over inputs $x \in \mathcal{X} \subset \mathbb{R}^d$ and labels $y \in \mathcal{Y}$. Heterogeneity across clients arises from discrepancies in the input marginals $P_k(x)$, corresponding to domain shift, and in the label marginals $P_k(y)$, corresponding to label imbalance. These two sources of heterogeneity jointly induce differences in the posterior distributions $P_k(y \mid x)$, leading to inconsistent local objectives

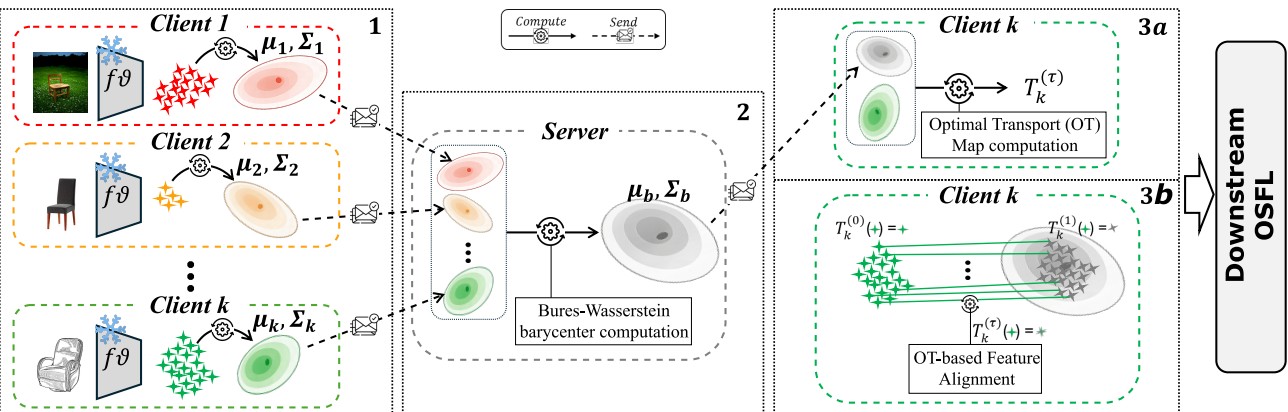

*Figure 1.* Overview of the SLOT-Align workflow. Clients compute first- and second-order feature statistics using a frozen encoder. The server aggregates them via a Bures–Wasserstein barycenter. Client-specific optimal transport maps are then computed and applied to align features before downstream OSFL.

and misaligned feature representations across clients.

All clients share a frozen encoder obtained via pretraining on an external dataset:

$$f_\theta : \mathcal{X} \to \mathbb{R}^m.$$

Given an input $x$, the encoder produces a representation $z = f_\theta(x)$. Notably, the use of a shared, frozen encoder ensures that all local input spaces are mapped into a common latent manifold in $\mathbb{R}^m$. This shared representation provides the necessary metric space upon which the pushforward measures can be directly compared, ensuring that observed discrepancies arise solely from statistical shifts rather than architectural divergence. The induced feature distribution at client $k$ is given by the pushforward of the input marginal, $Q_k := (f_\theta)_\# P_k(x)$, where $P_k(x)$ denotes the marginal of $P_k(x, y)$ on $\mathcal{X}$. That is, $Q_k$ is the distribution of $z = f_\theta(x)$ when $x \sim P_k(x)$. Since $P_k(x) = \sum_{y \in \mathcal{Y}} P_k(x \mid y) \, P_k(y)$, discrepancies among clients in both $P_k(x)$ and the label marginal $P_k(y)$ induce differences in the family $\{Q_k\}_{k=1}^K$, which generally differ in first- and second-order statistics (i.e., means and covariances), reflecting both domain-specific feature geometry and class-conditional biases.

Under OSFL constraints, the server cannot iteratively correct these discrepancies through repeated optimization or model exchange. As a result, mismatches among the feature distributions $Q_k$ directly affect the server-side training objective and cannot be mitigated during learning. This limitation is particularly severe when domain shift and label shift interact, as skewed class proportions distort the empirical feature moments contributed by each client in a non-uniform manner.

To enable principled alignment under these constraints, we adopt a statistics-based representation of client feature distributions within the 2-Wasserstein space $(\mathcal{P}_2(\mathbb{R}^m), W_2)$. Here, $\mathcal{P}_2(\mathbb{R}^m)$ denotes the space of probability measures on

$\mathbb{R}^m$ with finite second moment. The 2-Wasserstein distance between two distributions $Q_i, Q_j \in \mathcal{P}_2(\mathbb{R}^m)$ is defined as

$$W_2^2(Q_i, Q_j) = \inf_{\pi \in \Pi(Q_i, Q_j)} \int_{\mathbb{R}^m \times \mathbb{R}^m} \|z - z'\|_2^2 \, d\pi(z, z'),$$

where $\Pi(Q_i, Q_j)$ is the set of all couplings with marginals $Q_i$ and $Q_j$. This distance captures discrepancies in both location and geometric spread and induces displacement interpolations corresponding to continuous transport of probability mass. Since all client representations lie in a common latent metric space, these discrepancies can be interpreted as mass displacements in $\mathbb{R}^m$, and alignment amounts to transporting feature distributions toward a common reference. The quadratic optimal transport distance provides a canonical geometry-aware discrepancy that captures both location and shape differences and induces displacement interpolations interpreted as continuous feature deformations. Moreover, for Gaussian measures the $W_2$ geometry admits closed-form optimal transport maps and well-defined geodesics, and Gaussian distributions form a totally geodesic submanifold of $(\mathcal{P}_2(\mathbb{R}^m), W_2)$. As a result, heterogeneous feature distributions can be harmonized using only first- and second-order statistics. This yields a tractable and communication-efficient formulation that is well suited to the one-shot setting and serves as the foundation for the SLOT-Align framework described in the following sections.

### 3.2. Local Feature Statistics Extraction

Each client $k$ computes compact first- and second-order summaries of its local feature distribution produced by a shared, frozen, pre-trained encoder $f_\theta$. Given local samples $x \sim P_k(x)$, the client extracts feature representations $z = f_\theta(x)$ and estimates the empirical mean:

$$\mu_k = \mathbb{E}_{x \sim P_k(x)}[f_\theta(x)],$$

together with the sample covariance:

$$\widehat{\Sigma}_k = \mathbb{E}_{x \sim P_k(x)} \left[ (f_\theta(x) - \mu_k)(f_\theta(x) - \mu_k)^\top \right].$$

In high-dimensional regimes or when local sample sizes are limited, the sample covariance may be ill-conditioned or rank-deficient. To ensure numerical stability and compatibility with optimal transport geometry, we adopt a shrinkage-based estimator and define the regularized covariance as:

$$\Sigma_k = \lambda_k \widehat{\Sigma}_k + (1 - \lambda_k)\sigma_k^2 \mathrm{Id},$$

where $\lambda_k \in [0, 1]$ is the Ledoit–Wolf shrinkage coefficient and $\sigma_k^2 = \frac{1}{m}\mathrm{tr}(\widehat{\Sigma}_k)$ denotes the average eigenvalue (the shrinkage target). This construction guarantees $\Sigma_k \in \mathcal{S}_{++}^m$, the cone of symmetric positive definite matrices, a step required for the Bures–Wasserstein geometry to be used in the subsequent stages.

The resulting pair $(\mu_k, \Sigma_k)$ defines a Gaussian measure $\mathcal{N}(\mu_k, \Sigma_k)$ that serves as a tractable proxy for the induced feature distribution $Q_k = (f_\theta)_\# P_k$. Since deep feature distributions are not Gaussian in general, we do not require $Q_k$ itself to be Gaussian. Rather, the Gaussian proxy retains the first- and second-order structure of $Q_k$, capturing dominant discrepancies in location and spread induced by domain shift through variations in $P_k(x)$ and by label shift through the reweighting of class-conditional feature distributions $Q_k(z \mid y)$. This representation results in a compact summary suitable for communication under strict budget constraints (as discussed in Section 4.4), while enabling closed-form aggregation and alignment under optimal transport geometry.

### 3.3. Construction of a Global Reference via Bures–Wasserstein Barycenter

Given the local Gaussian summaries $\{(\mu_k, \Sigma_k)\}_{k=1}^K$ collected from clients, the server constructs a global reference distribution that captures a common structure across heterogeneous feature spaces. The reference mean is computed as a weighted average

$$\mu_b = \sum_{k=1}^K w_k \mu_k,$$

where the weights are defined as $w_k = n_k/N$, with $N = \sum_{j=1}^K n_j$ denoting the total number of samples across clients and $n_k$ the number of samples at client $k$.

To aggregate covariances, we adopt the Bures–Wasserstein barycenter. Specifically, given client covariances $\{\Sigma_k\}_{k=1}^K \subset \mathcal{S}_{++}^m$, the barycenter covariance $\Sigma_b$ is defined as the unique minimizer

$$\Sigma_b = \arg \min_{\Sigma \in \mathcal{S}_{++}^m} \sum_{k=1}^K w_k B^2(\Sigma, \Sigma_k),$$

where $B(\cdot, \cdot)$ denotes the Bures distance. For covariance matrices $\Sigma_1, \Sigma_2 \in \mathcal{S}_{++}^m$, the squared Bures distance admits the closed-form expression

$$B^2(\Sigma_1, \Sigma_2) = \mathrm{tr}\left( \Sigma_1 + \Sigma_2 - 2\big(\Sigma_1^{1/2}\Sigma_2\Sigma_1^{1/2}\big)^{1/2} \right),$$

which arises from the quadratic optimal transport cost between Gaussian measures and induces a Riemannian geometry on $\mathcal{S}_{++}^m$.

The barycenter covariance $\Sigma_b$ can be computed via a standard fixed-point iteration. Starting from an initial positive definite matrix $\Sigma^{(0)}$, for instance $\Sigma^{(0)} = \sum_{k=1}^K w_k \Sigma_k$, the update takes the form

$$\Sigma^{(t+1)} = \sum_{k=1}^K w_k \left( \left(\Sigma^{(t)}\right)^{1/2} \Sigma_k \left(\Sigma^{(t)}\right)^{1/2} \right)^{1/2}.$$

For shrinkage-regularized covariances $\Sigma_k \in \mathcal{S}_{++}^m$, this iteration converges reliably in practice and produces a well-defined barycenter that balances client-specific variability with global consistency. The resulting pair $(\mu_b, \Sigma_b)$ defines a Gaussian reference distribution that serves as a common geometric anchor for subsequent alignments. The server transmits these global statistics back to the clients, concluding a one-shot, non-iterative exchange of compact feature summaries.

### 3.4. Geodesic Optimal Transport Maps for Client Alignment

After receiving the global reference statistics, each client constructs a transport map that aligns its Gaussian approximation $\mathcal{N}(\mu_k, \Sigma_k)$ to the reference $\mathcal{N}(\mu_b, \Sigma_b)$. Under quadratic cost, the optimal transport between Gaussian measures is affine and admits a closed-form solution. In particular, the optimal map from $\mathcal{N}(\mu_k, \Sigma_k)$ to $\mathcal{N}(\mu_b, \Sigma_b)$ is given by

$$A_k = \Sigma_b^{1/2} \left( \Sigma_b^{1/2} \Sigma_k \Sigma_b^{1/2} \right)^{-1/2} \Sigma_b^{1/2},$$

$$b_k = \mu_b - A_k \mu_k,$$

and acts on feature representations as

$$T_k(z) = A_k z + b_k.$$

This map exactly transports the source Gaussian to the target Gaussian in 2-Wasserstein space and corrects discrepancies in both mean and covariance.

However, the Gaussian summaries $(\mu_k, \Sigma_k)$ are estimated from finite local samples and provide only a proxy for the generally non-Gaussian feature distribution $Q_k$. In this regime, applying the full transport may over-correct relative to the true $Q_k$, motivating a controlled interpolation toward the barycentric reference.

To control the strength of alignment, SLOT-Align introduces an interpolation parameter $\tau \in [0, 1]$ that modulates the extent of transport. We define the interpolated map $T_k^{(\tau)}$ as the displacement interpolation between the identity and the optimal transport map $T_k$,

$$T_k^{(\tau)} = (1 - \tau)\,\mathrm{Id} + \tau\,T_k,$$

where $T_k$ denotes the optimal transport map between $\mathcal{N}(\mu_k, \Sigma_k)$ and $\mathcal{N}(\mu_b, \Sigma_b)$. An explicit affine expression for $T_k^{(\tau)}$ and its derivation are given in Appendix A.

For $\tau = 0$, the transformation reduces to the identity map and no alignment is applied, while for $\tau = 1$, the full optimal transport map $T_k$ is recovered.

Applying $T_k^{(\tau)}$ to a Gaussian input preserves Gaussianity. In particular, the pushforward of $\mathcal{N}(\mu_k, \Sigma_k)$ through $T_k^{(\tau)}$ is $\mathcal{N}(\mu_\tau, \Sigma_\tau)$, where

$$\mu_\tau = (1 - \tau)\mu_k + \tau\mu_b$$

is the linearly interpolated mean, and

$$\Sigma_\tau = \big((1 - \tau)\mathrm{Id} + \tau A_k\big)\Sigma_k\big((1 - \tau)\mathrm{Id} + \tau A_k\big)^\top$$

is the covariance evolving smoothly along the interpolation (see Appendix A for additional details).

Since $T_k$ is the optimal transport map between the endpoint Gaussians, the family $\{\mathcal{N}(\mu_\tau, \Sigma_\tau)\}_{\tau \in [0,1]}$ coincides with the Wasserstein displacement interpolation (McCann, 1997) connecting $\mathcal{N}(\mu_k, \Sigma_k)$ and $\mathcal{N}(\mu_b, \Sigma_b)$, that is, the unique constant-speed geodesic in 2-Wasserstein space.

Because the set of Gaussian measures is totally geodesic under the $W_2$ geometry, all intermediate aligned proxy distributions remain Gaussian and evolve smoothly between the source and the global reference.

This geodesic structure yields an explicit contraction property in the Gaussian proxy space, with the distance to the barycentric reference decreasing linearly with the interpolation level. Let $G_k = \mathcal{N}(\mu_k, \Sigma_k)$, $G_b = \mathcal{N}(\mu_b, \Sigma_b)$, and $G_k^{(\tau)} = (T_k^{(\tau)})_\# G_k$. Since $G_k^{(\tau)}$ lies at interpolation level $\tau$ along the constant-speed $W_2$ geodesic from $G_k$ to $G_b$, it satisfies

$$W_2(G_k^{(\tau)}, G_b) = (1 - \tau)W_2(G_k, G_b).$$

Thus, $\tau$ directly controls the amount of proxy-level discrepancy removed with respect to the barycentric reference. This property characterizes the closed-form alignment step in the Gaussian proxy space.

In practice, $\tau$ is treated as a global hyperparameter controlling the trade-off between preserving local feature geometry and enforcing distributional alignment toward the barycentric reference. In this work, we employ a single value of $\tau$ across all clients and datasets, without relying on heterogeneity estimates or problem-specific prior knowledge. This design preserves the non-iterative nature of SLOT-Align while providing a simple and controlled mechanism to modulate alignment strength.

### 3.5. Transformation and Integration with OSFL Algorithms

After computing the client-specific alignment map $T_k^{(\tau)}$, each client applies this transformation to all local feature representations produced by the frozen encoder, yielding aligned features

$$z' = T_k^{(\tau)}(z).$$

This operation induces a transformed feature distribution $Q_k' = T_{k\#}^{(\tau)} Q_k$ whose first- and second-order statistics match the intermediate Gaussian $\mathcal{N}(\mu_\tau, \Sigma_\tau)$ defined in Section 3.4. As a result, cross-client discrepancies in feature geometry caused by heterogeneous $P_k(x)$ and amplified by biased $P_k(y)$ are reduced prior to any server-side training. The features $z'$ can be fed to any OSFL algorithm relying on pretrained encoders, for improved robustness to joint domain and label shift through geometry-aware distribution alignment.

SLOT-Align acts as a learning-free transformation that operates only in local feature space and does not alter the structure, objective, or optimization procedure of the downstream OSFL pipeline. It incurs negligible computational overhead (see Section 4.4) and remains agnostic to the specific downstream loss function or training protocol.

## 4. Experiments

We evaluate SLOT-Align in an OSFL setting to assess its effectiveness under heterogeneous data distributions. Our experimental design focuses on three aspects: (i) performance under *domain shift compounded by label shift*, which constitutes the most challenging OSFL regime; (ii) robustness across different frozen backbone architectures; and (iii) compatibility with existing OSFL algorithms without modifying their optimization procedures.

*Table 1.* Performance under joint domain and label shift ($\alpha = 0.1$). Top-1 accuracy is reported per domain, along with macro-averaged accuracy (mean) and standard deviation (std) across domains. FedAvg is reported as a multi-round reference.

| Method | Office-Home | | | | | | Digits | | | | | | DomainNet | | | | | | | |
|---|---|---|---|---|---|---|---|---|---|---|---|---|---|---|---|---|---|---|---|---|
| | A | C | P | R | mean | std | Mnist | Usps | Svhn | Synth | mean | std | Cl | In | Pa | Qu | Re | Sk | mean | std |
| FedAvg | 76.95 | 72.98 | 88.19 | 88.33 | 81.36 | 6.80 | 79.04 | 81.15 | 63.08 | 68.07 | 71.82 | 7.58 | 64.05 | 30.37 | 53.81 | 25.58 | 64.28 | 53.18 | 48.10 | 15.26 |
| O-FedAvg | 66.09 | 55.45 | 69.22 | 66.01 | 64.19 | 5.21 | 76.36 | 73.98 | 49.99 | 58.45 | 64.70 | 10.93 | 56.93 | 24.55 | 43.34 | 17.56 | 53.09 | 40.69 | 37.69 | 12.53 |
| +SLOT | 67.57 | 67.51 | 78.95 | 74.16 | **72.05**(+7.86) | 4.81 | 85.61 | 76.97 | 56.03 | 71.84 | **72.61**(+7.91) | 10.77 | 49.90 | 27.73 | 45.16 | 20.03 | 54.56 | 43.91 | **40.26**(+2.57) | 12.18 |
| FedCGS | 80.52 | 74.58 | 94.74 | 87.61 | 84.37 | 7.56 | 82.18 | 81.46 | 53.29 | 51.55 | 67.12 | 14.72 | 66.67 | 31.74 | 58.16 | 20.66 | 73.08 | 58.11 | 51.40 | 18.82 |
| +SLOT | 80.66 | 75.95 | 94.89 | 88.69 | **85.05**(+0.68) | 7.28 | 84.05 | 73.44 | 63.05 | 59.9 | **70.11**(+2.99) | 9.48 | 70.00 | 40.59 | 63.02 | 28.54 | 75.42 | 62.21 | **56.63**(+5.23) | 16.59 |
| FedPFT | 67.99 | 61.25 | 85.51 | 82.01 | 74.19 | 9.94 | 84.74 | 82.23 | 61.60 | 64.85 | 73.36 | 10.23 | 62.85 | 29.13 | 54.71 | 23.54 | 65.86 | 52.13 | 48.04 | 16.10 |
| +SLOT | 75.99 | 72.06 | 89.64 | 85.24 | **80.74**(+6.55) | 7.02 | 86.87 | 83.19 | 62.75 | 69.18 | **75.50**(+2.14) | 9.89 | 65.54 | 34.83 | 57.45 | 26.73 | 68.37 | 56.48 | **51.57**(+3.53) | 15.46 |

*Table 2.* Performance under stronger joint domain and label shift ($\alpha = 0.05$). Top-1 accuracy is reported per domain, along with macro-averaged accuracy (mean) and standard deviation (std) across domains.

| Method | Office-Home | | | | | | Digits | | | | | | DomainNet | | | | | | | |
|---|---|---|---|---|---|---|---|---|---|---|---|---|---|---|---|---|---|---|---|---|
| | A | C | P | R | mean | std | Mnist | Usps | Svhn | Synth | mean | std | Cl | In | Pa | Qu | Re | Sk | mean | std |
| O-FedAvg | 65.08 | 54.50 | 66.38 | 67.46 | 63.36 | 5.18 | 76.73 | 80.97 | 52.74 | 52.87 | 65.83 | 13.11 | 38.67 | 19.32 | 37.20 | 17.54 | 42.08 | 34.01 | 31.47 | 9.54 |
| +SLOT | 67.46 | 62.46 | 73.80 | 73.88 | **69.40**(+6.04) | 4.78 | 80.45 | 71.20 | 56.80 | 59.15 | **66.90**(+1.07) | 9.54 | 42.49 | 22.73 | 40.43 | 19.40 | 46.03 | 37.21 | **34.72**(+3.25) | 10.04 |
| FedCGS | 76.82 | 70.00 | 88.96 | 86.62 | 80.60 | 7.63 | 83.60 | 80.37 | 45.90 | 57.85 | 66.93 | 15.68 | 63.41 | 29.76 | 55.75 | 20.67 | 68.65 | 56.16 | 49.07 | 17.62 |
| +SLOT | 77.64 | 74.43 | 90.99 | 85.86 | **82.23**(+1.63) | 6.56 | 88.22 | 78.92 | 49.09 | 65.45 | **70.42**(+3.49) | 14.74 | 66.02 | 34.67 | 59.02 | 25.23 | 70.31 | 58.89 | **52.36**(+3.29) | 16.55 |
| FedPFT | 70.07 | 58.76 | 80.24 | 84.45 | 73.38 | 9.93 | 82.36 | 83.16 | 46.74 | 44.82 | 64.27 | 18.51 | 60.48 | 28.66 | 51.39 | 21.74 | 59.81 | 51.38 | 45.58 | 14.98 |
| +SLOT | 75.53 | 71.80 | 87.49 | 85.50 | **80.08**(+6.70) | 6.59 | 85.77 | 73.99 | 50.77 | 67.02 | **69.39**(+5.12) | 12.67 | 62.53 | 32.30 | 53.94 | 24.78 | 63.11 | 54.63 | **48.55**(+2.97) | 14.73 |

## 4.1. Experimental Setup

**Datasets and client configuration.** To assess the proposed geometry-aware alignment, we consider three standard benchmarks typically adopted in cross-domain settings: **Office-Home** (Venkateswara et al., 2017), **Digits** (LeCun et al., 2002; Hull, 2002; Netzer et al., 2011; Roy et al., 2018), and **DomainNet** (Peng et al., 2019). **Office-Home** includes four domains: Art (A), Clipart (C), Product (P), and Real World (R), each containing 65 classes. **Digits** includes four domains: Mnist, Usps, Svhn, and Synthetic Digits (Synth), each with 10 categories. **DomainNet** comprises six domains: Clipart (*Cl*), Infograph (*In*), Painting (*Pa*), Quickdraw (*Qu*), Real (*Re*), and Sketch (*Sk*), with a total of 345 categories. Each domain is treated as a distinct client, resulting in one client per domain and inducing explicit domain shift through differences in $\mathbb{P}_k(x)$. To induce label shift, client-specific label marginals $\mathbb{P}_k(y)$ are generated using a Dirichlet distribution with concentration parameter $\alpha = 0.1$. Samples of each class are then allocated across clients according to these proportions, resulting in heterogeneous label distributions across clients. Unless otherwise stated, all experiments use a frozen ViT-B/32 encoder pretrained with CLIP. Additional backbones are evaluated in ablation experiments.

**Baselines and evaluation protocol** We evaluate SLOT-Align by integrating it into the following OSFL baselines, which operate under a single communication round for model training: **O-FedAvg** (McMahan et al., 2017; Guha et al., 2019), the standard one-shot extension of FedAvg; **FedCGS** (Guan et al., 2025), a statistics-based OSFL method leveraging global feature summaries; **FedPFT** (Beitollahi et al., 2025), an OSFL approach built on pretrained feature representations. We additionally report results for multi-round FedAvg (McMahan et al., 2017), which

relaxes the one-shot constraint, serving as an upper bound. For each OSFL method, we report results with and without SLOT-Align applied as a preprocessing step, without modifying the downstream training procedure. Within each experimental setting, all methods use the same frozen encoder and extracted features, so the reported gains are not due to encoder differences. All other baseline-specific training settings follow the corresponding original implementations. Performance is evaluated on a held-out test set for each client. We report top-1 accuracy per domain, macro-averaged accuracy across domains, and the standard deviation across domains as a measure of consistency under heterogeneity. All results are averaged over five independent runs. Unless otherwise stated, the alignment strength is fixed to $\tau = 0.4$. Covariance matrices computed by SLOT-Align are regularized using the Ledoit–Wolf shrinkage estimator in all experiments.

## 4.2. Main Results under Joint Domain and Label Shift

Table 1 reports results under domain shift compounded by label shift ($\alpha = 0.1$), using the CLIP-pretrained ViT-B/32 encoder as specified in Section 4.1. This setting, adopted as the primary evaluation scenario, represents the most challenging OSFL regime, as discrepancies in both input distributions $\mathbb{P}_k(x)$ and label proportions $\mathbb{P}_k(y)$ jointly distort the feature statistics available for one-shot aggregation.

Across all datasets, incorporating SLOT-Align consistently improves the macro-averaged accuracy of the evaluated OSFL baselines. These improvements are generally accompanied by a reduction in cross-domain performance variability, indicating more consistent behavior across heterogeneous clients.

On Office-Home, SLOT-Align provides clear improvements

*Table 3.* Performance under stronger joint domain and label shift ($\alpha = 0.01$). Top-1 accuracy is reported per domain, along with macro-averaged accuracy (mean) and standard deviation (std) across domains.

| Method | Office-Home | | | | | | Digits | | | | | | DomainNet | | | | | | | |
|---|---|---|---|---|---|---|---|---|---|---|---|---|---|---|---|---|---|---|---|---|
| | A | C | P | R | mean | std | Mnist | Usps | Svhn | Synth | mean | std | Cl | In | Pa | Qu | Re | Sk | mean | std |
| O-FedAvg | 58.07 | 55.98 | 75.08 | 75.28 | 66.10 | 9.11 | 50.80 | 60.22 | 24.90 | 32.57 | 42.12 | 14.06 | 36.39 | 18.37 | 32.90 | 14.15 | 38.92 | 30.88 | 28.60 | 9.17 |
| +SLOT | 66.44 | 67.86 | 81.58 | 80.86 | **74.19** (+8.09) | 7.06 | 55.67 | 58.88 | 27.76 | 34.40 | **44.18** (+2.06) | 13.35 | 43.37 | 22.71 | 39.68 | 17.38 | 44.54 | 38.28 | **34.33** (+5.97) | 10.43 |
| FedCGS | 68.59 | 67.25 | 89.04 | 83.56 | 77.11 | 9.40 | 46.62 | 63.93 | 7.27 | 26.15 | 35.99 | 21.30 | 62.26 | 30.12 | 52.25 | 16.94 | 67.30 | 52.64 | 46.92 | 17.76 |
| +SLOT | 72.29 | 73.74 | 90.17 | 85.78 | **80.49** (+3.38) | 7.66 | 61.55 | 64.18 | 13.86 | 30.35 | **42.48** (+6.49) | 21.22 | 65.23 | 35.89 | 56.39 | 21.75 | 68.77 | 56.07 | **50.69** (+3.77) | 16.61 |
| FedPFT | 61.50 | 57.58 | 80.26 | 80.61 | 69.99 | 10.54 | 69.47 | 64.51 | 11.74 | 29.03 | 43.69 | 24.15 | 56.28 | 26.72 | 46.09 | 15.11 | 56.12 | 48.62 | 41.49 | 15.38 |
| +SLOT | 70.69 | 68.27 | 85.79 | 83.82 | **77.14** (+7.15) | 7.74 | 74.95 | 69.31 | 20.67 | 38.98 | **50.98** (+7.29) | 22.21 | 61.98 | 32.63 | 51.62 | 19.36 | 63.60 | 52.79 | **47.00** (+5.51) | 15.94 |

for O-FedAvg and FedPFT and reduces cross-domain variability. Consistent improvements are obtained for FedCGS, although smaller than those observed for the other baselines, in line with the strong performance of the method prior to alignment. Overall, these results indicate that feature-level alignment is most effective for one-shot methods whose aggregation is sensitive to cross-client distribution mismatch, while still providing gains when baseline performance is already high. On Digits, SLOT-Align improves all evaluated one-shot baselines, and the resulting alignment is also associated with reduced cross-domain variability. The largest gains are observed for O-FedAvg, with positive improvements also obtained for FedCGS and FedPFT. These results suggest that feature-level alignment contributes to stabilizing performance under the combined effects of domain and label shift in this setting. On DomainNet, which represents the most diverse and large-scale benchmark considered, SLOT-Align improves all one-shot baselines and reduces discrepancies across domains. These results show that the proposed alignment remains effective in challenging regimes where high domain diversity is coupled with additional distributional heterogeneity.

## 4.3. Ablations and Analysis

**Stronger label skew.** The main results in Table 1 use $\alpha = 0.1$ to induce label shift. To assess robustness to stronger label imbalance, we further evaluate SLOT-Align with $\alpha = 0.05$ and $\alpha = 0.01$, where lower values of $\alpha$ induce more concentrated client label marginals. As reported in Tables 2 and 3, SLOT-Align consistently improves all evaluated OSFL baselines across Office-Home, Digits, and DomainNet. The improvements under both moderate and severe label skew indicate that SLOT-Align does not depend on a single heterogeneity configuration and remains robust when domain shift is compounded by increasingly imbalanced client label distributions.

**Additional backbones.** To assess robustness with respect to representation quality, we evaluate SLOT-Align using ResNet-18 pretrained on ImageNet (RN18) and ViT-B/32 pretrained on ImageNet (ViT32) on the Office-Home dataset. As shown in Table 6, SLOT-Align consistently improves all evaluated OSFL baselines across both backbones. While absolute performance varies with representation quality, the

relative gains remain stable, indicating that SLOT-Align is agnostic to the specific pretrained backbone.

*Table 4.* Domain-only shift on Office-Home with balanced label distributions. Top-1 accuracy is reported per domain, along with macro-averaged accuracy (mean) and standard deviation (std) across domains.

| Method | A | C | P | R | mean | std |
|---|---|---|---|---|---|---|
| FedAvg | 87.74 | 87.05 | 96.92 | 93.94 | 91.30 | 4.16 |
| O-FedAvg | 88.23 | 85.31 | 96.32 | 93.88 | 90.94 | 4.38 |
| +SLOT | 88.29 | 85.47 | 96.47 | 93.79 | 91.00 (+0.06) | 4.35 |
| FedCGS | 83.40 | 84.27 | 95.65 | 93.12 | 89.11 | 5.36 |
| +SLOT | 83.95 | 84.20 | 95.35 | 93.12 | 89.15 (+0.04) | 5.14 |
| FedPFT | 85.79 | 85.88 | 96.17 | 93.04 | 90.22 | 4.52 |
| +SLOT | 86.42 | 85.53 | 95.57 | 93.18 | 90.17 (-0.05) | 4.30 |

**Domain-only shift.** In this work, we target an OSFL setting characterized by domain shift, with additional label shift across clients. To contextualize our approach under the more general domain-shift-only setting, we perform an ablation in which label distributions are balanced across clients. In this setting, OSFL baselines already achieve strong performance. As shown in Table 4, SLOT-Align achieves performance on par with these baselines, with marginal gains or slight variations depending on the method. This behavior is consistent with the fact that, in the absence of label shift, the sample mean and covariance of client features provide a more faithful approximation of a shared class-conditional feature structure, reducing the need for alignment. Importantly, SLOT-Align does not introduce systematic degradation in this regime.

*Table 5.* Ablation of alignment strength $\tau$ on Office-Home under joint domain and label shift ($\alpha = 0.1$). Macro-averaged accuracy (mean) and standard deviation (std) across domains are reported.

| | Method | | | | | |
|---|---|---|---|---|---|---|
| | O-FedAvg + SLOT | | FedCGS + SLOT | | FedPFT + SLOT | |
| $\tau$ | mean | std | mean | std | mean | std |
| 0.0 | 64.19 | 5.21 | 84.37 | 7.56 | 74.19 | 9.94 |
| 0.2 | 68.11 | 4.38 | 84.40 | 7.62 | 78.17 | 7.72 |
| 0.4 | **72.05** | 4.81 | **85.05** | 7.28 | **80.74** | 7.02 |
| 0.6 | 71.25 | 5.05 | 83.88 | 7.77 | 80.36 | 7.57 |
| 0.8 | 71.06 | 5.61 | 81.76 | 7.89 | 78.91 | 7.11 |
| 1.0 | 69.42 | 5.70 | 80.05 | 8.44 | 75.36 | 7.33 |

**Alignment strength $\tau$.** Table 5 analyzes the sensitivity of SLOT-Align to the interpolation parameter $\tau$ on Office-Home under domain shift, with additional label shift. Set-

*Table 6.* Frozen backbones on Office-Home under joint domain and label shift. Top-1 accuracy is reported per domain, with macro-averaged accuracy (mean) and standard deviation (std).

| Backbone | Method | A | C | P | R | mean | std |
|---|---|---|---|---|---|---|---|
| | FedAvg | 63.97 | 50.20 | 73.10 | 70.67 | 64.17 | 8.90 |
| ViT32 | O-FedAvg | 58.00 | 39.08 | 61.34 | 61.83 | 55.06 | 9.34 |
| | +SLOT | 56.82 | 45.22 | 65.23 | 64.53 | **57.95** (+2.89) | 8.05 |
| | FedCGS | 63.51 | 49.85 | 73.72 | 71.64 | 64.68 | 9.38 |
| | +SLOT | 64.06 | 51.15 | 76.95 | 73.39 | **66.39** (+1.71) | 9.98 |
| | FedPFT | 62.28 | 47.20 | 71.57 | 70.11 | 62.79 | 9.67 |
| | +SLOT | 62.83 | 50.76 | 72.75 | 69.95 | **64.07** (+1.28) | 8.49 |
| | FedAvg | 46.41 | 36.13 | 61.04 | 55.86 | 49.62 | 9.51 |
| RN18 | O-FedAvg | 25.76 | 18.66 | 33.59 | 30.23 | 27.06 | 5.59 |
| | +SLOT | 29.03 | 22.08 | 35.45 | 34.20 | **30.19** (+3.13) | 5.27 |
| | FedCGS | 48.42 | 34.96 | 67.42 | 60.63 | 52.86 | 12.37 |
| | +SLOT | 49.93 | 41.76 | 70.27 | 62.08 | **56.01** (+3.15) | 10.96 |
| | FedPFT | 44.22 | 31.83 | 60.56 | 56.27 | 48.22 | 11.20 |
| | +SLOT | 45.59 | 37.35 | 62.04 | 57.49 | **50.62** (+2.40) | 9.73 |

ting $\tau = 0.0$ corresponds to no alignment, effectively recovering the baseline without applying SLOT-Align, while increasing $\tau$ progressively enforces transport toward the global reference. Across all baselines, intermediate values of $\tau$ lead to consistent accuracy improvements and reduced cross-domain variability. In contrast, very small values provide limited benefit, while larger values tend to degrade performance, indicating that overly strong alignment can be detrimental. These results support the use of a moderate alignment strength and motivate the choice $\tau = 0.4$ adopted in the main experiments, which provides a stable trade-off across methods without dataset-specific tuning. Additional PCA visualizations illustrating the qualitative effect of $\tau$ are reported in Appendix B.

### 4.4. Computational and Communication Complexity

SLOT-Align operates as a lightweight preprocessing layer on feature representations already produced by existing OSFL pipelines based on frozen pretrained encoders. On the client side, the only additional computation is the estimation of first- and second-order feature statistics, which for a client with $n_k$ samples and feature dimension $m$ requires $\mathcal{O}(n_k m)$ time and $\mathcal{O}(m^2)$ memory. On the server side, SLOT-Align aggregates the received statistics and computes a Bures–Wasserstein barycenter in $\mathcal{S}_{++}^m$. Each fixed-point iteration has cost $\mathcal{O}(m^3)$, and convergence is typically achieved in a small number of iterations for shrinkage-regularized covariances. Since this computation depends only on the feature dimensionality, it is independent of dataset size. The corresponding affine transport maps are obtained in closed form and incur negligible additional cost.

From a communication perspective, SLOT-Align relies exclusively on the exchange of the local sample count $n_k$, the mean $\mu_k \in \mathbb{R}^m$ and the covariance $\Sigma_k \in \mathbb{R}^{m \times m}$, resulting in a per-client communication cost of $\mathcal{O}(m + m^2)$. For

example, with float16 communication and $m = 512$, transmission requires about 264 KB per client using symmetric packing, since $\Sigma_k$ is symmetric. No raw data, gradients, model parameters, logits, class-wise summaries, or synthetic samples are transmitted. Although we do not claim formal privacy guarantees, SLOT-Align is compatible with standard privacy-enhancing mechanisms, such as secure aggregation. Additional experiments with finer client partitions, reported in Appendix C, show that SLOT-Align remains beneficial under a larger number of clients, with unchanged per-client communication requirements.

## 5. Conclusion

In this paper, we introduced SLOT-Align, a learning-free, geometry-aware alignment mechanism that operates on compact feature statistics and corrects distributional discrepancies prior to downstream OSFL training. We showed that, under one-shot constraints, the joint effect of domain shift in $P_k(x)$ and label shift in $P_k(y)$ induces misaligned feature distributions that cannot be reliably addressed by existing OSFL methods alone. SLOT-Align leverages closed-form optimal transport for Gaussian measures to harmonize client representations using a single exchange of statistics, without modifying downstream optimization procedures. Extensive experiments across multiple datasets, backbone architectures, and state-of-the-art OSFL baselines demonstrate consistent performance gains and improved robustness, particularly in challenging heterogeneous settings.

## Impact Statement

This paper presents work whose goal is to advance the field of Machine Learning. There are many potential societal consequences of our work, none of which we feel must be specifically highlighted here.

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

# A. Displacement Interpolation and Gaussian Wasserstein Geodesics

In this appendix we derive the explicit affine form of the interpolated transport map used in Section 3.4 and recall its connection with Wasserstein displacement interpolation and Gaussian geodesics.

Let $\mathcal{N}(\mu_k, \Sigma_k)$ and $\mathcal{N}(\mu_b, \Sigma_b)$ be two nondegenerate Gaussian measures on $\mathbb{R}^m$. It is well known (Gelbrich, 1990; Villani, 2008) that the optimal transport map between them is affine and given by

$$T_k(z) = A_k z + b_k, \qquad b_k = \mu_b - A_k \mu_k, \tag{1}$$

where

$$A_k = \Sigma_b^{1/2} \left( \Sigma_b^{1/2} \Sigma_k \Sigma_b^{1/2} \right)^{-1/2} \Sigma_b^{1/2}. \tag{2}$$

Following the definition of displacement interpolation (McCann, 1997), the constant-speed Wasserstein geodesic between $\mathcal{N}(\mu_k, \Sigma_k)$ and $\mathcal{N}(\mu_b, \Sigma_b)$ is obtained by interpolating between the identity map and the optimal transport map,

$$T_k^{(\tau)} = (1 - \tau)\operatorname{Id} + \tau\, T_k, \qquad \tau \in [0, 1], \tag{3}$$

and by defining the interpolated measures $\nu_\tau := \left( T_k^{(\tau)} \right)_\# \mathcal{N}(\mu_k, \Sigma_k)$.

Since $T_k$ is affine, the interpolated map is also affine and can be written as

$$T_k^{(\tau)}(z) = \left[ (1-\tau)\operatorname{Id} + \tau A_k \right] z + \tau(\mu_b - A_k \mu_k). \tag{4}$$

To make explicit the evolution of the mean and covariance along the interpolation, it is convenient to rewrite the translation term as

$$\tau(\mu_b - A_k \mu_k) = (1 - \tau)\mu_k + \tau\mu_b - \left[ (1 - \tau)\operatorname{Id} + \tau A_k \right]\mu_k. \tag{5}$$

This yields the equivalent centered form

$$T_k^{(\tau)}(z) = \mu_\tau + \left[ (1-\tau)\operatorname{Id} + \tau A_k \right](z - \mu_k), \qquad \mu_\tau = (1 - \tau)\mu_k + \tau\mu_b. \tag{6}$$

In this representation, the map transports deviations from the source mean $z - \mu_k$ by the linear operator $(1 - \tau)\operatorname{Id} + \tau A_k$ and anchors them at the interpolated mean $\mu_\tau$. In particular,

$$T_k^{(\tau)}(\mu_k) = \mu_\tau, \tag{7}$$

so the mean evolves linearly along the interpolation, and for $Z \sim \mathcal{N}(\mu_k, \Sigma_k)$ the covariance evolves as

$$\Sigma_\tau = \operatorname{Cov}\!\left( T_k^{(\tau)}(Z) \right) = \left[ (1-\tau)\operatorname{Id} + \tau A_k \right]\Sigma_k \left[ (1-\tau)\operatorname{Id} + \tau A_k \right]^\top. \tag{8}$$

It is a classical result that the family $\{\mathcal{N}(\mu_\tau, \Sigma_\tau)\}_{\tau \in [0,1]}$ coincides with the unique constant-speed Wasserstein geodesic between $\mathcal{N}(\mu_k, \Sigma_k)$ and $\mathcal{N}(\mu_b, \Sigma_b)$ (Gelbrich, 1990; McCann, 1997). Moreover, the set of Gaussian measures is totally geodesic in $(\mathcal{P}_2(\mathbb{R}^d), W_2)$, so that all intermediate distributions along the geodesic remain Gaussian (Takatsu, 2011).

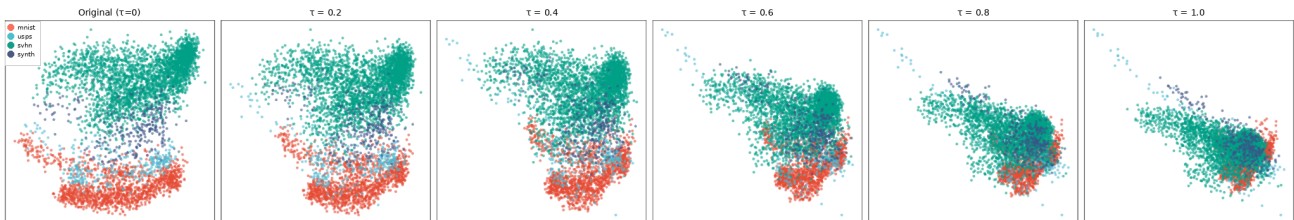

*Figure 2.* PCA projections of Digits feature representations for increasing values of $\tau$, colored by domain. A subset of classes is used for readability.

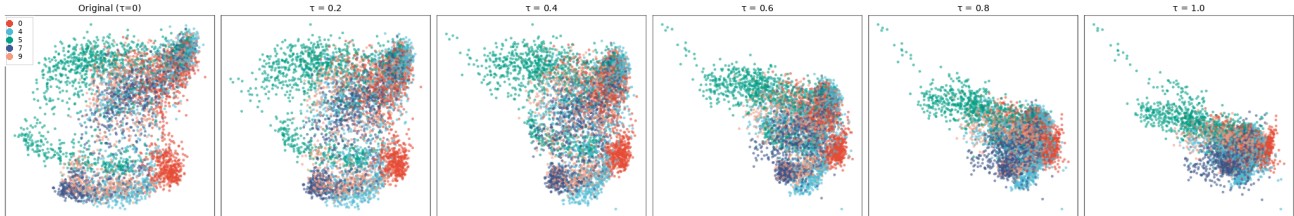

*Figure 3.* PCA projections of Digits feature representations for increasing values of $\tau$, colored by class. A subset of classes is used for readability.

## B. Qualitative Effect of the Alignment Strength

To complement the quantitative ablation in Table 5, we visualize the effect of the interpolation parameter $\tau$ on the aligned feature representations. For readability, we use a subset of the Digits benchmark containing half of the classes, which provides a larger number of samples per class and makes the PCA projections easier to interpret. Figures 2 and 3 show PCA projections of the features for increasing values of $\tau$, colored by domain and by class, respectively. In the domain-colored view, increasing $\tau$ progressively reduces cross-domain discrepancies, showing that the transport moves client representations toward a common reference. In the class-colored view, however, overly strong transport tends to over-compress the representation around the barycentric reference, potentially reducing class separation. These qualitative trends are consistent with the quantitative behavior observed in Table 5: intermediate values of $\tau$, corresponding to intermediate points along the Wasserstein geodesic, provide the best compromise between reducing domain mismatch and preserving discriminative structure.

## C. Scalability

The main experiments follow a standard federated evaluation protocol in which each domain is treated as one client, consistent with prior federated learning work on domain shift (Li et al., 2021b; Son et al., 2024; Chen et al., 2023b). To further assess the behavior of SLOT-Align under a larger client count, we introduce a finer partitioning protocol in which each original domain is split into two clients, thereby doubling the total number of clients while preserving the underlying domain structure. Table 7 reports results with O-FedAvg under domain shift compounded by label shift ($\alpha = 0.1$). SLOT-Align improves macro-averaged accuracy across Office-Home, Digits, and DomainNet, indicating that the proposed alignment remains beneficial when client partitions become more granular. The communication structure remains unchanged: each client transmits one pair of aggregate feature statistics ($\mu_k, \Sigma_k$), and the server computes a single Bures–Wasserstein barycentric reference before broadcasting the alignment information.

*Table 7.* Scalability with finer client partitions under domain shift compounded by label shift ($\alpha = 0.1$), with $K = 8$ clients for Office-Home and Digits and $K = 12$ clients for DomainNet. Results report macro-averaged accuracy (mean) and standard deviation (std) across clients.

| Method | Office-Home (K = 8) | | Digits (K = 8) | | DomainNet (K = 12) | |
|---|---|---|---|---|---|---|
| | mean | std | mean | std | mean | std |
| O-FedAvg | 60.73 | 6.05 | 56.79 | 19.01 | 37.67 | 11.78 |
| +SLOT | **62.06** (+1.33) | 4.78 | **60.51** (+3.72) | 17.11 | **40.51** (+2.84) | 11.78 |

