# OpenReview forum: "Distribution Alignment for One-Shot Federated Learning via Optimal Transport"
_ICML.cc/2026/Conference — ICML 2026 regular_

### Official Review · Reviewer_Nm4U · 2026-03-10

**Soundness:** 4
**Presentation:** 3
**Significance:** 4
**Originality:** 3
**Overall Recommendation:** 5
**Confidence:** 2

**Summary:**

This paper addresses the issue of inconsistent client feature distributions in OSFL, arising from both domain drift and label drift. It proposes SLOT-Align, a lightweight feature alignment framework based on optimal transport. The approach enables geometric alignment of local features by having clients transmit first- and second-order feature statistics. On the server side, these statistics are used to construct a Bures-Wasserstein centre of mass as a global reference. This reference is then returned to clients to construct a closed-form optimal transport mapping. The entire process requires no training, no auxiliary data, and can be seamlessly integrated into existing OSFL methods that rely on pre-trained encoders. Experiments across multiple cross-domain datasets validate the method's effectiveness.

**Compliance With Llm Reviewing Policy:**

Affirmed.

**Final Justification:**

The author has addressed my concerns.

**Key Questions For Authors:**

1) The scope of the paper's contributions requires clarification
2) The experimental client count is too low, falling short of real-world large-scale FL scenarios; expansion to large-scale client experiments is needed
3) In-depth analysis of τ's sensitivity is required

**Limitations:**

yes

**Strengths And Weaknesses:**

Strengths:
1) SLOT-Align is straightforward and effective, with strong compatibility
2) Comprehensive experiments yielding outstanding results

Weaknesses:
1) The scope of the paper's contributions requires clarification
2) The experimental client count is too low, falling short of real-world large-scale FL scenarios; expansion to large-scale client experiments is needed
3) In-depth analysis of τ's sensitivity is required

---

> ### Author Rebuttal · Authors · 2026-03-29
>
> We thank the reviewer for the feedback and address the points below.
>
> > **The scope of the paper's contributions requires clarification.**
>
> We agree that the scope should be stated more explicitly. Our proposed SLOT-Align is a new, learning-free preprocessing module for OSFL methods. Its role is to correct cross-client feature misalignment induced by **domain shift when compounded by additional label shift** before downstream one-shot training. Our main contribution is therefore a **geometry-aware alignment mechanism** that is compatible with existing OSFL pipelines, requires only a single exchange of compact statistics, and leaves the downstream optimization procedure unchanged.
>
> > **The experimental client count is too low and falls short of large-scale FL scenarios.**
>
> Our original setup follows the standard protocol used in multi-domain federated learning benchmarks, where each domain is treated as one client, consistent with prior FL works studying domain shift in this setting [1-3]. A truly large-scale FL study is outside the scope of this paper and is also not well supported by these benchmarks, which provide only a limited number of domains. However, to assess scalability, we add an additional experiment with a finer client partition on each dataset (Office-Home, Digits, DomainNet; α=0.1), obtained by further splitting each original domain into multiple clients, thereby **doubling the total number of clients**. SLOT-Align remains beneficial on all three datasets:
>
> - Office-Home:
> |Method|C1|C2|C3|C4|C5|C6|C7|C8|mean|std|
> |-|-|-|-|-|-|-|-|-|-|-|
> |O-FedAvg|61.52|63.99|58.09|48.00|59.41|71.08|61.15|62.62|60.73|6.05|
> |+SLOT|60.51|60.66|58.36|53.94|59.82|73.64|62.58|66.97|**62.06 (+1.33)**|4.78|
>
> - Digits:
> |Method|C1|C2|C3|C4|C5|C6|C7|C8|mean|std|
> |-|-|-|-|-|-|-|-|-|-|-|
> |O-FedAvg|43.55|89.83|25.56|72.57|63.80|43.30|66.99|48.75|56.79|19.01|
> |+SLOT|61.05|89.79|28.17|73.71|67.13|48.57|63.44|52.23|**60.51 (+3.72)**|17.11|
>
> - DomainNet:
> |Method|C1|C2|C3|C4|C5|C6|C7|C8|C9|C10|C11|C12|mean|std|
> |-|-|-|-|-|-|-|-|-|-|-|-|-|-|-|
> |O-FedAvg|48.75|42.44|25.11|23.84|45.30|42.36|19.56|20.19|54.54|50.40|38.74|40.82|37.67|11.78|
> |+SLOT|51.72|45.78|28.10|26.94|48.55|45.37|21.34|22.50|55.91|51.97|43.28|44.69|**40.51 (+2.84)**|11.78|
>
> These results suggest that the method remains effective under finer client partitions. To further clarify the scalability point, SLOT-Align does not introduce extra communication rounds as the number of clients grows: each client still sends one compact pair $(\mu_k,\Sigma_k)$, and the server computes a single global barycentric reference before broadcasting the alignment parameters. Thus, **the communication pattern of SLOT-Align remains unchanged as the client count increases**. The finer-partition experiments are also meant to support this structural scalability.
>
>     [1] Li, Xiaoxiao, et al. “FedBN: Federated Learning on Non-IID Features via Local Batch Normalization.” International Conference on Learning Representations, 2021.
>     [2] Son, Ha Min, et al. "Feduv: Uniformity and variance for heterogeneous federated learning." Proceedings of the IEEE/CVF conference on computer vision and pattern recognition. 2024.
>     [3] Chen, Haokun, et al. "Fraug: Tackling federated learning with non-iid features via representation augmentation." Proceedings of the IEEE/CVF international conference on computer vision. 2023.
>
> > **A more in-depth analysis of $\tau$ sensitivity is required.**
>
> We already report a $\tau$ ablation in the paper (Section 4.3, Table 3), showing that intermediate values consistently outperform both no alignment ($\tau=0$) and full transport ($\tau=1$), which motivates our choice of a single fixed moderate value $\tau=0.4$ across methods, datasets, and backbones. We strengthen this analysis by adding PCA visualizations for increasing $\tau$, which show the progressive displacement interpolation induced by the transport map.
>
> Here are the links to the PCA visualizations colored by [domain](https://ibb.co/Cp7kcXqG) and by [class](https://ibb.co/5gtBfgkm). These plots make the role of $\tau$ more interpretable. In the domain-colored view, increasing $\tau$ progressively reduces cross-domain discrepancies, showing that the alignment acts in the intended direction. In the class-colored view, however, overly strong transport tends to over-compress the representation toward the common reference, which could reduce class separation and thus harm discriminative structure. This qualitatively supports the empirical pattern in Table 3: intermediate values of $\tau$ (i.e. intermediate points along the Wasserstein geodesic) provide the best compromise between correcting domain mismatch and preserving useful class geometry.

---

> > ### Author Rebuttal · Reviewer_Nm4U · 2026-04-04
> >
> > Thank you for your reply, and I will improve my rating.

---

> > > ### Author Response · Authors · 2026-04-05
> > >
> > > Thank you for the positive acknowledgment and for updating your rating. Many thanks for your feedback and consideration.

---

### Official Review · Reviewer_vpZh · 2026-03-12

**Soundness:** 3
**Presentation:** 3
**Significance:** 2
**Originality:** 2
**Overall Recommendation:** 4
**Confidence:** 3

**Summary:**

This paper tackles the data heterogeneity challenge in One-Shot Federated Learning (OSFL) by proposing the method SLOT-Align, which aims to align client feature distributions before downstream OSFL training using optimal transport. Experiments on Office-Home, Digits, and DomainNet have been performed and the proposed method shows conssitent performance improvements.

**Compliance With Llm Reviewing Policy:**

Affirmed.

**Final Justification:**

My concerns have been addressed, and I would like to keep my rating as positive.

**Key Questions For Authors:**

Please see the weakness.

**Limitations:**

yes

**Strengths And Weaknesses:**

## Strength
- The paper tackle the challenging heterogeneity issue and identifies the specific challenges in OSFL.
- The idea is clear to follow and the method design shows good compatibility with existing methods.
- The paper is overall well organized.


## Weakness
- The proposed method approximates deep feature distributions with Gaussian models, if the feature distributions are multi-modal or heavily skewed, aligning based on first and second moments could be misleading. In addition, a visualization of feature distributions before and after alignment could further strengthen this work.

- While the ablation on \tau is appreciated, using a single fixed value for all clients and datasets is a strong simplification. Client heterogeneity can vary significantly, and clients with different shift level may requires varying alignment level.

- The method requires sharing statistics, in high-dimensional feature spaces, covariance matrices can potentially leak information about the underlying data distribution, which raise potential privacy concerns.

- The evaluation on label shift is limited, it is necessary to perform label-shift evaluations with different skew-levels.

---

> ### Author Rebuttal · Authors · 2026-03-29
>
> We thank the reviewer for the positive assessment and constructive suggestions.
>
> > **Approximating deep feature distributions with Gaussians may be misleading if features are multi-modal or skewed. Visualization ...**
>
> We acknowledge that deep feature distribution are not exactly Gaussian in general. In SLOT-Align, the Gaussian model is a tractable proxy that enables closed-form, one-shot transport in feature space, rather than an exact density assumption. To better support this point, we added PCA visualizations before and after alignment, and for increasing $\tau$, showing the progressive displacement interpolation induced by the transport map. These plots show that the alignment evolves smoothly in a controlled way, rather than through an abrupt distortion. Here are the links to the PCA visualizations colored by [domain](https://ibb.co/Cp7kcXqG) and by [class](https://ibb.co/5gtBfgkm).
>
> > **Single fixed $\tau$; Different clients may require different alignment strength.**
>
> This is a fair point. Our use of a single global $\tau$ was motivated by the one-shot setting, where avoiding client-specific tuning, heterogeneity estimation, and extra communication is particularly important. Empirically, a moderate value ($\tau = 0.4$) was stable across all datasets, backbones, and downstream OSFL methods we considered. Nevertheless, while we agree that client-adaptive $\tau$ is a promising extension, we believe a fixed $\tau$ design is a meaningful practical result on its own, because it shows that SLOT-Align can provide consistent gains under a single shared setting, without requiring per-client calibration.
>
> > **Sharing covariance matrices in high-dimensional feature spaces may raise privacy concerns.**
>
> We agree that this deserves discussion. SLOT-Align shares only **dataset-level aggregate statistics** in a frozen feature space, not raw data, per-sample features, gradients, logits, **class-wise summaries**, or synthetic samples. This makes the transmitted information substantially coarser than in many OSFL alternatives. At the same time, aggregate statistics might still reveal some information about the underlying distribution, so we do not claim formal privacy guarantees. We will clarify this point and note that the method is compatible with standard privacy-enhancing mechanisms, such as secure aggregation.
>
> > **Limited label shift evaluation; test different skew levels.**
>
> We perform additional experiments under stronger label skew, using different Dirichlet coefficients α=0.05 and α=0.01 in addition to α=0.1, across all benchmarks and baselines. The results consistently show that SLOT-Align remains beneficial as skew increases.
>
> - Office-Home (α=0.05, α=0.01):
>
> |Method|A|C|P|R|mean|std|
> |-|-|-|-|-|-|-|
> O-FedAvg|65.08|54.50|66.38|67.46|63.36|5.18
> +SLOT|67.46|62.46|73.80|73.88|**69.40 (+6.04)**|4.78
> FedCGS|76.82| 70.00|88.96| 86.62|80.60| 7.63
> +SLOT|77.64|74.43|90.99|85.86|**82.23 (+1.63)**|6.56
> FedPFT|70.07|58.76|80.24|84.45|73.38|9.93
> +SLOT|75.53|71.80|87.49|85.50|**80.08 (+6.70)**|6.59
>
> |Method|A|C|P|R|mean|std|
> |-|-|-|-|-|-|-|
> O-FedAvg|58.07|55.98|75.08|75.28|66.1|9.11
> +SLOT|66.44|67.86|81.58|80.86|**74.19 (+8.09)**|7.06
> FedCGS|68.59|67.25|89.04|83.56|77.11| 9.4
> +SLOT|72.29|73.74|90.17|85.78|**80.49 (+3.38)**|7.66
> FedPFT|61.5|57.58|80.26|80.61|69.99| 10.54
> +SLOT|70.69|68.27|85.79|83.82|**77.14 (+7.15)**|7.74
>
> - Digits (α=0.05, α=0.01)
>
> |Method|Mnist|Usps|Svhn|Synth|mean|std|
> |-|-|-|-|-|-|-|
> O-FedAvg|76.73|80.97|52.74|52.87|65.83|13.11
> +SLOT|80.45|71.20|56.80|59.15|**66.90 (+1.07)**|9.54
> FedCGS|83.60|80.37|45.90|57.85|66.93|15.68
> +SLOT|88.22|78.92|49.09|65.45|**70.42 (+3.49)**|14.74
> FedPFT|82.36|83.16|46.74|44.82|64.27|18.51
> +SLOT|85.77|73.99|50.77|67.02|**69.39 (+5.12)**|12.67
>
> |Method|Mnist|Usps|Svhn|Synth|mean|std|
> |-|-|-|-|-|-|-|
> O-FedAvg|50.80|60.22|24.90|32.57|42.12|14.06
> +SLOT|55.67|58.88|27.76|34.40|**44.18 (+2.06)**|13.35
> FedCGS|46.62|63.93|7.27|26.15|35.99|21.30
> +SLOT|61.55|64.18|13.86|30.35|**42.48 (+6.49)**|21.22
> FedPFT|69.47|64.51|11.74|29.03|43.69|24.15
> +SLOT|74.95|69.31|20.67|38.98|**50.98 (+7.29)**|22.21
>
> - DomainNet (α=0.05, α=0.01):
>
> |Method|Cl|In|Pa|Qu|Re|Sk|mean|std|
> |-|-|-|-|-|-|-|-|-|
> O-FedAvg|38.67|19.32|37.20|17.54|42.08|34.01|31.47|9.54
> +SLOT|42.49|22.73|40.43|19.40|46.03|37.21|**34.72 (+3.25)**|10.04
> FedCGS|63.41|29.76|55.75|20.67|68.65|56.16|49.07|17.62
> +SLOT|66.02|34.67|59.02|25.23|70.31|58.89|**52.36 (+3.29)**|16.55
> FedPFT|60.48|28.66|51.39|21.74|59.81|51.38|45.58|14.98
> +SLOT|62.53|32.30|53.94|24.78|63.11|54.63|**48.55 (+2.97)**|14.73
>
> |Method|Cl|In|Pa|Qu|Re|Sk|mean|std|
> |-|-|-|-|-|-|-|-|-|
> O-FedAvg|36.39|18.37|32.90|14.15|38.92|30.88|28.60|9.17
> +SLOT|43.37|22.71|39.68|17.38|44.54|38.28|**34.33 (+5.97)**|10.43
> FedCGS|62.26|30.12|52.25|16.94|67.30|52.64|46.92|17.76
> +SLOT|65.23|35.89|56.39|21.75|68.77|56.07|**50.69 (+3.77)**|16.61
> FedPFT|56.28|26.72|46.09|15.11|56.12|48.62|41.49|15.38
> +SLOT|61.98|32.63|51.62|19.36|63.60|52.79|**47.00 (+5.51)**|15.94

---

> > ### Author Rebuttal · Reviewer_vpZh · 2026-04-03
> >
> > Thanks for the authors response, I have no more questions and would like to keep my current score.

---

> > > ### Author Response · Authors · 2026-04-05
> > >
> > > Thank you for the positive acknowledgment and for confirming that our response addressed your concerns. We appreciate your time and consideration.

---

### Official Review · Reviewer_GrTK · 2026-03-12

**Soundness:** 3
**Presentation:** 3
**Significance:** 2
**Originality:** 2
**Overall Recommendation:** 3
**Confidence:** 3

**Summary:**

The paper studies an interesting problem in one-shot federated learning and proposes a simple alignment module for handling feature misalignment under heterogeneous client distributions. The idea is intuitive, the technical design is relatively clean, and the empirical results are generally encouraging. I also appreciate the plug-in nature of the proposed method, which makes it potentially applicable to existing one-shot FL pipelines in a relatively flexible manner.

**Compliance With Llm Reviewing Policy:**

Affirmed.

**Final Justification:**

I will retain my original score.

**Key Questions For Authors:**

Please see the weaknesses.

**Limitations:**

yes

**Strengths And Weaknesses:**

Pros
1.The paper addresses a meaningful and relevant problem in one-shot federated learning, namely feature misalignment under heterogeneous client distributions.
2.The plug-in design is appealing. The proposed module can be integrated with existing one-shot FL pipelines without substantially changing their core optimization procedures.
3.The proposed idea is intuitive and technically clean. The overall method is easy to follow, and the formulation is relatively well structured.

Cons：
1.The method relies on strong empirical assumptions. Real feature distributions are often non-Gaussian, and it is unclear whether first- and second-order moments are sufficient to preserve class-conditional and discriminative structure.
2.It is unclear whether Wasserstein geometry is truly necessary. The paper does not compare against simpler second-order moment matching methods, so the source of the improvement remains uncertain.
3.The method introduces extra overhead. Although it remains one-shot, it adds local statistics extraction, server-side barycenter computation, and client-side transport.
4.The heterogeneity setting is limited. Experiments mainly focus on a single label-skew setting, making robustness under different heterogeneity levels unclear.
5.Scalability is not sufficiently analyzed. It remains unclear how the method performs with a larger number of clients or in more realistic multi-client settings.
6.The presentation can be further improved. The manuscript contains several formatting inconsistencies, missing punctuation marks, and occasionally non-standard academic expressions, which somewhat weaken the clarity and overall presentation quality.

---

> ### Author Rebuttal · Authors · 2026-03-29
>
> We thank the reviewer for the feedback. We address each point below and provide additional results.
>
> > **1. Moment matching may not preserve class discriminative structure.**
>
> In our method, Gaussian summaries are a tractable proxy that enables a closed-form formulation, one-shot transport map, rather than an exact modeling claim, allowing to get significant improvements over the baselines.
>
> To further show that this approximation does not harm discriminative structure, we added:
> 1) **PCA visualizations** for increasing $\tau$, showing smooth displacement interpolation rather than abrupt deformation. Here are the links to the PCA visualizations colored by [class](https://ibb.co/5gtBfgkm) and by [domain](https://ibb.co/Cp7kcXqG);
>
> 2) a **non-federated linear-probe (LP) test** on each full dataset, comparing the same linear classifier trained on original (LP-Orig) vs. aligned frozen features (LP-Align). The improvement suggest that SLOT-Align preserves class-conditional and downstream discriminative structure.
>
> - Office-Home:
> |Method|A|C|P|R|mean|std|
> |-|-|-|-|-|-|-|
> LP-Orig|76.54|70.35|90.87|88.43|81.31|8.44
> LP-Align|81.62|76.99|94.67|89.53|**85.47 (+4.16)**|6.85
>
> - Digits:
> |Method|Mnist|Usps|Svhn|Synth|mean|std|
> |-|-|-|-|-|-|-|
> LP-Orig|78.81|82.31|62.18|58.7|69.65|10.25
> LP-Align|84.87|85.77|62.59|64.95|**73.61 (+3.96)**|10.85
>
> - DomainNet:
> |Method|Cl|In|Pa|Qu|Re|Sk|mean|std|
> |-|-|-|-|-|-|-|-|-|
> LP-Orig|63.85|29.15|53.52|23.55|65.46|51.93|47.69|16.10
> LP-Align|63.28|32.95|55.16|24.62|65.52|54.15|**49.02 (+1.33)**|15.24
>
> > **2. Moment-matching methods comparison.**
>
> We added comparisons against simpler moment-matching alternatives:
> 1) **mean-only alignment** (μ-only),
> 2) **covariance-only alignment** (Σ-only),
> 3) **Euclidean whitening-recoloring** (CORAL).
>
> |Method|A|C|P|R|mean|std|
> |-|-|-|-|-|-|-|
> O-FedAvg|66.09|55.45|69.22|66.01|64.19|5.21
> +μ-only|65.25|57.76|70.87|66.67|65.14|4.73
> +Σ-only|67.76|64.00|75.14|72.42|69.83|4.27
> +CORAL|59.84|53.73|64.62|58.06|59.06|3.91
> +SLOT|66.64|65.56|76.23|72.63|**70.26 (+6.07)**|4.37
>
> These comparisons suggest that the improvement is not explained by simpler second-order moment matching alone. CORAL underperforms, while SLOT-Align outperforms all alternatives, empirically supporting the benefit of combining moment matching with the Wasserstein/Bures geometry.
>
> > **3. Method overhead.**
>
> SLOT-Align adds a lightweight preprocessing module based on feature statistics extraction, barycentric reference computation, and an affine feature transform. It does not introduce iterative communication, extra local training rounds, distillation, or synthetic data generation. The asymptotic cost is already reported in the paper, and we will clarify it more explicitly in the revision to better contextualize this overhead relative to heavier OSFL methods.
>
> > **4. Heterogeneity.**
>
> We add experiments with stronger label skew, using different Dirichlet parameters (α=0.05 and α=0.01) other than α=0.1. The results show that SLOT-Align remains beneficial as heterogeneity increases. Below we report results on Office-Home:
>
> - α=0.05
> |Method|A|C|P|R|mean|std|
> |-|-|-|-|-|-|-|
> O-FedAvg|65.08|54.50|66.38|67.46|63.36|5.18
> +SLOT|67.46|62.46|73.80|73.88|**69.40 (+6.04)**|4.78
> FedCGS|76.82| 70.00|88.96| 86.62|80.60|7.63
> +SLOT|77.64|74.43|90.99|85.86|**82.23 (+1.63)**|6.56
> FedPFT|70.07|58.76|80.24|84.45|73.38|9.93
> +SLOT|75.53|71.80|87.49|85.50|**80.08 (+6.70)**|6.59
>
> - α=0.01
> |Method|A|C|P|R|mean|std|
> |-|-|-|-|-|-|-|
> O-FedAvg|58.07|55.98|75.08|75.28|66.1|9.11
> +SLOT|66.44|67.86|81.58|80.86|**74.19 (+8.09)**|7.06
> FedCGS|68.59|67.25|89.04|83.56|77.11| 9.4
> +SLOT|72.29|73.74|90.17|85.78|**80.49 (+3.38)**|7.66
> FedPFT|61.5|57.58|80.26|80.61|69.99| 10.54
> +SLOT|70.69|68.27|85.79|83.82|**77.14 (+7.15)**|7.74
>
> The same trend is observed on the other benchmarks under both additional skew settings (α=0.05, 0.01): on Digits, SLOT-Align improves all the baselines, with gains up to +6.64; on DomainNet, it again improves all the baselines, with gains up to +5.97. Due to space limitations, we report here only the Office-Home tables, the corresponding **Digits and DomainNet tables are included in our response to Reviewer vpZh**.
>
> > **5. Scalability.**
>
> Our original setup follows the standard protocol for the multi-domain benchmarks used, where each domain is treated as one client, consistent with prior OSFL and FL work tackling domain-shift. To further assess scalability, we added across all datasets a finer partition that doubles the number of clients. SLOT-Align remains beneficial, improving mean accuracy with O-FedAvg by +1.33 on Office-Home, +3.72 on Digits, and +2.84 on DomainNet. The communication pattern remains unchanged, since each client still sends only compact first- and second-order statistics. For space limitations, we reported the summary here, **full tables are included in the rebuttal to Reviewer Nm4U**.
>
> > **6. Presentation**
>
> Thanks for pointing it out. We will revise formatting, punctuation, and wording.

---

> > ### Author Rebuttal · Reviewer_GrTK · 2026-04-02
> >
> > Thank you for your response. My main concerns have been addressed. However, the additional supplementary experiments have raised new concerns regarding the rationality of the experimental design in the manuscript.

---

### Official Review · Reviewer_wFyR · 2026-03-13

**Soundness:** 2
**Presentation:** 3
**Significance:** 3
**Originality:** 3
**Overall Recommendation:** 4
**Confidence:** 3

**Summary:**

This paper addresses the problem of One-Shot Federated Learning, where clients communicate with the server only once, precluding iterative optimization. The key challenge lies in handling heterogeneity across clients, particularly domain shift and label shift. The authors propose SLOT-Align, a geometry-aware feature alignment framework that operates as a preprocessing layer on frozen encoder representations. The method computes compact first- and second-order feature statistics locally using a shared frozen encoder, aggregates statistics via Burès-Wasserstein barycenter to create a global reference distribution, constructs client-specific optimal transport maps to align local feature distributions toward the reference, applies interpolated transport maps with a trainable parameter to control alignment strength. The framework claims to be computationally efficient and compatible with any downstream OSFL algorithm, requiring only a single exchange of statistics without modifying OSFL pipelines.

**Compliance With Llm Reviewing Policy:**

Affirmed.

**Final Justification:**

My concerns are solved, and I raised my score as promised.

**Key Questions For Authors:**

Refer to Weakness.

**Limitations:**

yes

**Strengths And Weaknesses:**

## Strengths

1. The paper clearly articulates the challenge of joint domain and label shift in OSFL settings. The formalization as a distribution alignment problem using 2-Wasserstein distance is principled and appropriate.

2. The use of optimal transport theory is well-justified. The closed-form solutions for Gaussian optimal transport maps and the connection to Wasserstein geodesics provide theoretical elegance.

3. SLOT-Align operates cleanly as a modular preprocessing layer requiring only (1) local statistics computation, (2) single server aggregation, and (3) local optimal transport map construction. This compatibility with frozen encoders and existing OSFL methods is a genuine practical advantage over learning-based distillation approaches.

4. $O(mn)$ client-side computation and $O(m^3)$ server-side Wasserstein barycenter computation are reported. The independence of computational cost from dataset size is a valuable property for federated settings.

5. Figure 1 effectively communicates the workflow. Mathematical notation is generally consistent and well-defined (though see weaknesses).

6. The paper acknowledges both domain shift and label shift, and demonstrates via Table 2 that the method performs comparably under domain-only shift, indicating the alignment gains are robust when both sources of heterogeneity are present.


## Weaknesses

1. Missing theoretical analysis and convergence guarantees: No formal analysis of how feature distribution alignment in the frozen encoder space translates to improved downstream task performance. No convergence rates or sample complexity bounds for the overall OSFL procedure. The paper claims SLOT-Align is "learning-free," but this is misleading: alignment strength $\tau$ is still a global hyperparameter requiring tuning (though Table 3 suggests τ = 0.4 is robust).

2.  No theoretical justification for why 2-Wasserstein distance is the appropriate metric for this problem versus alternatives (e.g., maximum mean discrepancy, KL divergence)

3. Missing comparisons with recent OSFL methods. Several cited recent works (FedDEO, FedSDC, FeDBayes) are not compared against despite being contemporary OSFL approaches

4. Some baseline implementations may differ from original papers (e.g., different encoder architectures); unclear if all baselines use identical ViT-B/32 encoders

5. FedAvg is included as an upper bound (multi-round), but computational budgets differ fundamentally which is not a fair comparison

6. The paper claims to address label shift, but the mechanism is unclear.

7. Missing ablation isolating the contribution of second-order statistics (covariance) vs. first-order statistics (mean)

8. Gaussian approximation assumptions insufficiently justified. The method assumes feature distributions are well-approximated by Gaussians, but frozen encoder features from modern vision models (ViT-B, ResNet-18) may not be Gaussian. For non-Gaussian distributions, Wasserstein optimal transport maps could be significantly misspecified.

---

> ### Author Rebuttal · Authors · 2026-03-29
>
> We thank the reviewer for the feedback. We address the main points below.
>
> > **1. Missing theoretical analysis / convergence ; “learning-free” wording**
>
>
> Despite $\tau$ might be learned, we remark that the method studied in this paper is training-free: we fix $\tau=0.4$ throughout all experiments and report an ablation to assess sensitivity to this hyperparameter, rather than tuning it per method, dataset, or client.
> We do not provide convergence or sample-complexity guarantees for the full OSFL pipeline, since SLOT-Align is a closed-form preprocessing alignment step applied before downstream training while leaving the OSFL pipeline unchanged. What can be stated formally is the behavior of the alignment step in the Gaussian proxy space. Let $G_k=\mathcal N(\mu_k,\Sigma_k)$ and $G_b=\mathcal N(\mu_b,\Sigma_b)$. If $G_k^{(\tau)} := (T_k^{(\tau)})_{\sharp} G_k$, then $G_k^{(\tau)}$ lies on the unique constant-speed $W_2$ geodesic from $G_k$ to $G_b$, hence $W_2(G_k^{(\tau)},G_b)=(1-\tau)W_2(G_k,G_b).$
> Thus, $\tau$ gives an exact contraction of discrepancy in the proxy geometry. We will add this proposition and clarify that our theory concerns the alignment module, not end-to-end OSFL pipeline.
>
> > **2. $W_2$ instead of KL/MMD**
>
> Our goal is not only to measure discrepancy, but to obtain in one shot (i) a global reference, (ii) an explicit client-side map, and (iii) a controlled partial-alignment path. In the Gaussian proxy space, $W_2$ provides all three in closed form: the Bures-Wasserstein barycenter, the affine OT map, and the geodesic interpolation through $\tau$. KL and MMD can measure mismatch, but they do not naturally provide this barycenter-map-geodesic structure. Thus, our claim is not that $W_2$ is universally optimal, but that it is the natural tractable choice for this one-shot alignment formulation.
>
> > **3. Missing comparisons with more OSFL methods**
>
> Our comparisons focus on OSFL pipelines that are directly compatible with SLOT-Align, namely methods based on **shared frozen encoders / feature statistics**, since SLOT-Align is a modular preprocessing layer for that regime. Methods such as FedDEO and FedSD2C rely on substantially different generative / synthetic-data pipelines, while FeDBayes addresses a different setting.
>
> > **4. Encoders consistency**
>
> We will clarify this. Baselines follow the original repositories; in the main experiments, all compared baselines use the same frozen ViT-B/32 encoder, and in ablations all methods again use the same backbone within each experiment. Thus, the gains are not due to encoder differences.
>
> > **5. FedAvg is not a fair one-shot comparison**
>
> We agree. Multi-round FedAvg is included only as a **reference point** to contextualize the gap between strict OSFL and iterative FL. It is **not** intended as a fair one-shot comparator, and our conclusions do not rely on that comparison.
>
> > **6. Claim about label shift**
>
> We agree that the wording should be more precise. SLOT-Align is **not** a label-shift correction method. Rather, it mitigates **feature misalignment induced by domain shift when compounded by label shift**, by reducing the moment-level feature mismatch that becomes more severe under skewed client label marginals. We will revise the wording accordingly.
>
> > **7. Ablation on first- vs second-order..**
>
> We added explicit ablations for **mean-only** (μ-only), **covariance-only** (Σ-only), and a simpler second-order Euclidean alternative (**CORAL**):
>
> Method|A|C|P|R|mean|std
> -|-|-|-|-|-|-
> O-FedAvg|66.09|55.45|69.22|66.01|64.19|5.21
> +μ-only|65.25|57.76|70.87|66.67|65.14|4.73
> +Σ-only|67.76|64.00|75.14|72.42|69.83|4.27
> +CORAL|59.84|53.73|64.62|58.06|59.06|3.91
> +SLOT|66.64|65.56|76.23|72.63|**70.26 (+6.07)**|4.37
>
> These comparisons suggest that second-order alignment drives most of the gain, but the alignment geometry matters: CORAL underperforms substantially, while full SLOT-Align achieves the best result.
>
> > **8. Gaussian approximation**
>
> We agree that frozen encoder features are not Gaussian in general, and we will state this more explicitly. SLOT-Align does **not** assume that the true latent distribution $Q_k=(f_\theta)_{\sharp}P_k$ is Gaussian. It aligns a **Gaussian proxy** $\mathcal N(\mu_k,\Sigma_k)$ defined by transmitted first- and second-order statistics, i.e., the **moment-level geometry** rather than the exact full distributions. For strongly multimodal features, the affine OT map is an approximation, which is precisely what motivates the interpolated map $T_k^{(\tau)}=(1-\tau)\mathrm{Id}+\tau T_k$. Empirically, intermediate $\tau$ outperform $\tau=1$, consistent with reducing dominant moment mismatch while avoiding over-correction. We also add PCA visualizations (here, colored by [class](https://ibb.co/5gtBfgkm) and by [domain](https://ibb.co/Cp7kcXqG)) showing that increasing $\tau$ progressively reduces cross-domain discrepancy, while full transport can over-compress class structure; intermediate points provide the best compromise.

---

> > ### Author Rebuttal · Reviewer_wFyR · 2026-04-04
> >
> > The rebuttal adequately addresses most of my initial concerns, I would raise my score in the final justification

---

> > > ### Author Response · Authors · 2026-04-05
> > >
> > > Thank you again for your careful review, we are glad that the rebuttal addressed most of your initial concerns.
> > >
> > > We also appreciate your note that you would raise your score in the final justification. If appropriate, we would be grateful if your score could be updated accordingly before the end of the review process.
> > >
> > > Thanks very much for your time.

---

### Decision · Program_Chairs · 2026-04-30

**Decision:**

Accept (regular)

**Comment:**

**Summary as I observed.**
In this paper, the authors propose SLOT-Align, a geometry-aware feature alignment framework for one-shot federated learning under heterogeneous client distributions, which harmonizes local representations using optimal transport with a single round of communication. The effectiveness of the proposed method is validated through extensive experiments across multiple benchmarks and settings. The paper receives final scores of 5/4/4/3.

**Strengths as I observed.**
The reviewers generally agree that the paper addresses an important and practically relevant challenge in one-shot federated learning, namely feature misalignment caused by heterogeneous client distributions, which is consistent with my assessment. The proposed optimal-transport-based alignment strategy is technically clean and well motivated, and its plug-in design makes it easy to integrate with existing OSFL frameworks without modifying their training procedures. Moreover, the empirical evaluation is conducted across multiple datasets and heterogeneity settings and shows consistent performance gains. The paper is also clearly written and generally well organized, which helps make the technical ideas accessible.

**Weaknesses as I observed.**
Some reviewers raised concerns regarding the Gaussian approximation used to model feature distributions, the necessity of the Wasserstein geometry compared with simpler alternatives, and the limited analysis of scalability and heterogeneity settings. In addition, questions were raised about the sharing of feature statistics and the sensitivity of the alignment strength parameter. In the rebuttal, the authors responded to these concerns with additional experiments and clarifications and largely addressed the reviewers' questions. I suggest that the final version further clarify the role of the Gaussian proxy assumption, strengthen the justification of the optimal transport formulation, and expand the discussion of scalability and privacy considerations.

After carefully reviewing the original paper, the reviewers' comments, and the authors' responses, I believe that the strengths of this paper outweigh its weaknesses, and the proposed SLOT-Align framework provides a meaningful contribution to distribution alignment for one-shot federated learning. Given that, I recommend acceptance.